# Single-cell analysis of two severe COVID-19 patients reveals a monocyte-associated and tocilizumab-responding cytokine storm

Chuang Guo [1,8], Bin Li[1,8], Huan Ma[1], Xiaofang Wang[2], Pengfei Cai[1], Qiaoni Yu[1], Lin Zhu[1], Liying Jin[1], Chen Jiang[1], Jingwen Fang[3], Qian Liu[1], Dandan Zong[1], Wen Zhang[1], Yichen Lu[1], Kun Li [1], Xuyuan Gao[1], Binqing Fu[1,4], Lianxin Liu[2], Xiaoling Ma[5], Jianping Weng[6], Haiming Wei [1,4], Tengchuan Jin [1,4✉], Jun Lin [1,4✉] & Kun Qu [1,4,7✉]

Several studies show that the immunosuppressive drugs targeting the interleukin-6 (IL-6) receptor, including tocilizumab, ameliorate lethal inflammatory responses in COVID-19 patients infected with SARS-CoV-2. Here, by employing single-cell analysis of the immune cell composition of two severe-stage COVID-19 patients prior to and following tocilizumab-induced remission, we identify a monocyte subpopulation that contributes to the inflammatory cytokine storms. Furthermore, although tocilizumab treatment attenuates the inflammation, immune cells, including plasma B cells and CD8[+] T cells, still exhibit robust humoral and cellular antiviral immune responses. Thus, in addition to providing a high-dimensional dataset on the immune cell distribution at multiple stages of the COVID-19, our work also provides insights into the therapeutic effects of tocilizumab, and identifies potential target cell populations for treating COVID-19-related cytokine storms.

[1] Department of Oncology, The First Affiliated Hospital of USTC, Division of Molecular Medicine, Hefei National Laboratory for Physical Sciences at Microscale, School of Basic Medical Sciences, Division of Life Sciences and Medicine, University of Science and Technology of China, 230021 Hefei, Anhui, China. [2] Department of Hepatobiliary Surgery, the First Affiliated Hospital of USTC, Division of Life Sciences and Medicine, University of Science and Technology of China, 230021 Hefei, Anhui, China. [3] HanGene Biotech, Xiaoshan Innovation Polis, 31200 Hangzhou, Zhejiang, China. [4] CAS Center for Excellence in Molecular Cell Sciences, the CAS Key Laboratory of Innate Immunity and Chronic Disease, University of Science and Technology of China, 230027 Hefei, Anhui, China. [5] Department of Laboratory Medicine, The First Affiliated Hospital of USTC, Division of Life Sciences and Medicine, University of Science and Technology of China, 230001 Hefei, Anhui, China. [6] Department of Endocrinology and Metabolism, The First Affiliated Hospital of USTC, Division of Life Sciences of Medicine, University of Science and Technology of China, 230026 Hefei, China. [7] School of Data Science, University of Science and Technology of China, 230026 Hefei, China. [8] These authors contributed equally: Chuang Guo, Bin Li. ✉email: jint@ustc.edu.cn; linjun7@ustc.edu.cn; qukun@ustc.edu.cn

A s of May 1, 2020, the WHO has reported 224,172 deaths among 3,175,207 confirmed cases of infection with severe acute respiratory syndrome coronavirus 2 (SARS-CoV-2), and these numbers are still growing rapidly[1]. Approximately 14% of patients with COVID-19 experienced severe disease, and 5% were critically ill, among whom there was a 49% fatality rate[2]; it has been speculated that this high mortality rate is related to abnormal immune system activation[3–5]. Hence, there is an urgent need for researchers to understand how the immune system responds to SARS-CoV-2 viral infection at the severe stage, which may highlight potential effective treatment strategies.

Studies have shown that the inflammatory storm caused by excessive immune responses was associated with mortality in COVID-19[6,7]. Plasma concentrations of several inflammatory cytokines, such as granulocyte-macrophage colony-stimulating factor (GM-CSF), interleukin (IL)-6[4], tumour necrosis factor α (TNF-α), IL-2, 7, 10, and granulocyte colony-stimulating factor (G-CSF)[8], were increased after SARS-CoV-2 infection. Further investigation demonstrated that peripheral inflammatory monocytes and pathogenic T cells may induce cytokine storms in severe COVID-19 patients[4,6]. Tocilizumab, an immunosuppressive drug that targets IL-6 receptors, has been used to treat severe COVID-19 patients[9,10], as it is effective for treating severe and even life-threatening cytokine-release syndrome[11,12]. After receiving tocilizumab, the body temperature of patients returned to normal after 24 h, and tocilizumab was shown to significantly decrease the concentration of oxygen inhalation of COVID-19 patients by the 5th day of treatment[13]. Despite the apparent efficacy of tocilizumab for treating severe COVID-19 patients, the lack of single-cell-level analyses has prevented any deepening of the understanding of how tocilizumab impacts the typical COVID-19-induced activation of an inflammatory storm.

In the present study, we profile the single-cell transcriptomes of 13,239 peripheral blood mononuclear cells (PBMCs) isolated at the severe and remission disease stages from 2 severe COVID-19 patients treated with tocilizumab. We identify a severe stage-specific monocyte subpopulation that contributes to the inflammatory cytokine storm in patients. Tocilizumab treatment weakens the inflammatory response while sustaining humoral and cellular antiviral immune responses in posttreatment COVID-19 patients. Our study thus provides a high-resolution dataset on the immune context at multiple stages of COVID-19 and helps to explain how a promising candidate drug alters immune cell populations and reduces patient mortality.

## Results

**An atlas of peripheral immune cells in COVID-19 patients**. We obtained five peripheral blood samples from two severe COVID-19 patients at three different time points, including the severe and remission stages, during tocilizumab treatment (Fig. 1a). Specifically, we collected blood samples on day 1 within 12 h of tocilizumab administration and on day 5 for both patients; we also obtained a blood sample from patient P2 on day 7 of tocilizumab treatment because P2 still had a positive result for the SARS-CoV-2 nucleic acid test of a throat swab specimen on day 5. On day 1, the patients both had a decreased number of lymphocytes based on healthy reference intervals as well as increased percentages of neutrophils, elevated concentrations of C-reactive protein and increased expression of IL-6 (Supplementary Data 1). Since the clinical symptoms of most of the severe COVID-19 patients, including both patients in this study, were remarkably improved after 5 days of tocilizumab treatment[13] (Supplementary Data 1), we defined the blood draws after day 5 as occurring in the remission stage. For patient P2, we obtained another blood draw on day 7, when his nucleic acid test became negative (Fig. 1a). It is worth

noting that patient P1 was discharged on day 8, and patient P2 was discharged on day 10, and these discharged patients were both negative at 3 days after a nucleic acid test of a throat swab specimen.

We isolated PBMCs from the COVID-19 patient blood samples and subjected them to single-cell mRNA sequencing (scRNA-seq) using the 10X platform (Fig. 1a). After rigorous quality control (QC) definition (Supplementary Fig. 1a–d, Supplementary Data 2), low-quality cells were filtered; we also removed cell doublets using Scrublet[14]. Correlation of the gene expression in the samples from either patient emphasized the excellent reproducibility between the technical and biological replicates of our dataset (Supplementary Fig. 1e, f). After QC and doublet removal, our dataset comprised a total of 13,239 high-quality transcriptomes from single PBMCs.

Due to the similarities between the single-cell transcriptomes of most of the identified cell subsets at the severe and remission disease stages, we initially combined the samples from both patients from day 1, which was the severe stage, and those from from day 5 (and day 7 for P2), which was the remission stage; we also conducted separate analyses for each patient, which yielded similar data trends (Supplementary Fig. 2a, b). In total, the combined analyses of all the single-cell transcriptomes for the COVID-19 patients included 4344 cells from the severe disease stage and 8895 from the remission disease stage.

To investigate the heterogeneity among the PBMCs from the COVID-19 patients compared to that in PBMCs from healthy controls, we applied Seurat[15] (version 3.1.4) to integrate our COVID-19 single-cell transcriptomes with the published single-cell profiles of healthy PBMCs from the 10X official website[16], enabling analysis of a total of 68,190 cells (see "Methods"). We then normalized and clustered the gene expression matrix; this identified 18 unique cell subsets, which were visualized via uniform manifold approximation and projection (UMAP) (Fig. 1b–d). The cell lineages of monocytes, CD4$^+$ and CD8$^+$ T cells, γδT cells, natural killer (NK) cells, B cells, plasma B cells, myeloid dendritic cells (mDCs), plasmacytoid dendritic cells (pDCs), platelets, and CD34$^+$ progenitor cells were identified based on the expression of known marker genes (Fig. 1e). This analysis represents the delineation of the landscape of circulating immune cells from severe COVID-19 patients.

We also used another integration method, Harmony[17], to help assess the accuracy of the cell clustering results from Seurat[15] (version 3.1.4) and again visualized the results with UMAP (Supplementary Fig. 3a). We found extensive similarities between the identified cell subsets and the detected gene expression patterns in the cell clusters with the two integration methods (Supplementary Fig. 3b, c), supporting the robustness of our cell clustering results.

We next explored the distribution of immune cells from the severe- and remission-stage COVID-19 patients as well as healthy control individuals (Supplementary Fig. 4a). We observed that a number of subpopulations, such as pDCs (cluster 15), mDCs (cluster 10), and most monocytes (clusters 2 and 13) were present in remission-stage COVID-19 patients and in healthy controls but not in severe COVID-19 patients (Supplementary Fig. 4b), indicating that tocilizumab treatment gradually restores the normal distribution of these cell types in circulating blood. Some cell subsets such as NK cells (cluster 7) and CD4$^+$ T cells (clusters 1 and 4) were quite heterogeneous between the two COVID-19 patients, so we did not examine these cell types further. These analyses revealed the conspicuous presence of four cell populations that were uniquely present in the COVID-19 patients (albeit to different extents in the severe vs. remission disease stages), including a monocyte subpopulation (cluster 9), plasma B cells (cluster 11), effector CD8$^+$ T cells (cluster 6), and proliferative MKI67$^+$CD8$^+$ T cells (cluster 12) (Supplementary Fig. 4c). Given

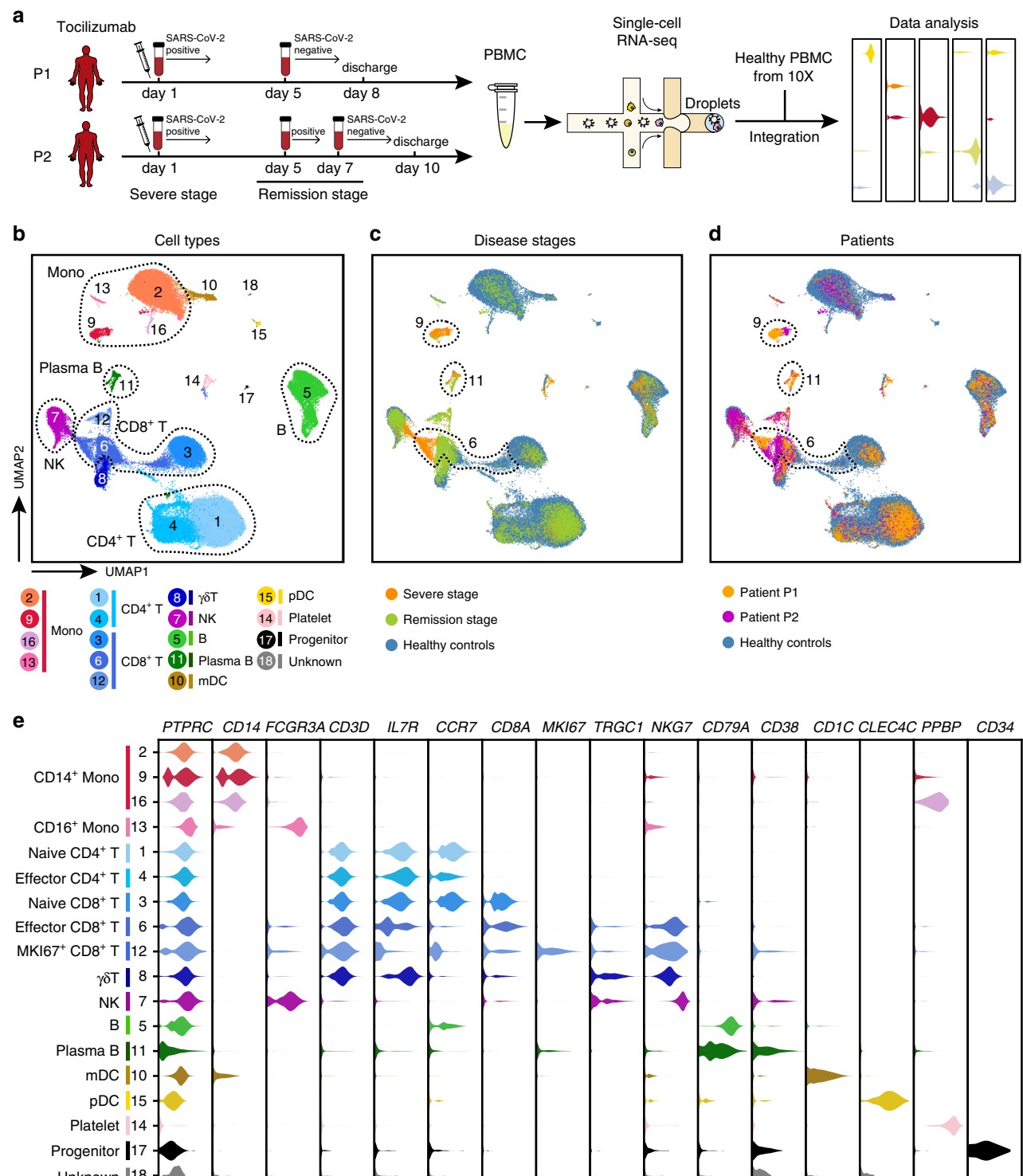

**Fig. 1 An atlas of peripheral immune cells in severe COVID-19 patients. a** Flowchart depicting the overall design of the study. Blood draws from patient P1 were performed at 2 time points (day 1 and day 5) and from P2 at 3 time points (day 1, day 5 and day 7). P1 on day 1 and P2 on day 1 and day 5 were positive based on a nucleic acid test of a throat swab specimen. P1 on day 5 and P2 on day 7 were negative based on a nucleic acid test of a throat swab specimen. Patients on day 1 were at the severe stage, and patients were in the remission stage on day 5 (P1 and P2); the day 7 blood draw for P2 (still in the remission stage) was performed based on a positive nucleic acid test on day 5. Note that the samples on day 1 were collected within 12 hours of tocilizumab treatment. **b**-**d** UMAP representations of integrated single-cell transcriptomes of 69,237 PBMCs, with 13,239 cells derived from our COVID-19 patients and 55,998 derived from the 10X Genomics official website[16]. Cells are colour-coded by clusters (**b**), disease state (**c**), and sample origin (**d**). Dotted circles represent cell types with a > 5% proportion within PBMCs in (**b**), and clusters significantly enriched in patients versus controls are shown in (**c**, **d**). Mono, monocyte; NK, natural killer cells; mDCs, myeloid dendritic cells; pDCs, plasmacytoid dendritic cells. **e** Violin plots of selected marker genes (upper row) for multiple cell subpopulations. The left column presents the cell subtypes identified based on combinations of marker genes.

the aim of our study of characterizing the COVID-19-specific and tocilizumab-sensitive immune cell populations of COVID-19 patients, the majority of our subsequent detailed analyses focused on these four cell populations.

**A monocyte subpopulation contributes to the cytokine storm.** Monocytes have been reported to play a vital role in CAR-T-induced cytokine-release syndrome[18] and in SARS-CoV-2 infection-triggered inflammatory storms[4], so we explored the features and functions of the aforementioned monocyte subpopulation that we detected in our single-cell analysis of the two COVID-19 patients. We detected 1,677 monocytes in the patients, with 916 detected in the severe disease stage and 761 detected in the remission stage; we examined the data from these monocyte alongside the data for 9517 monocytes from healthy controls. The UMAP plot displayed two main clusters of monocytes that were clearly segregated (Fig. 2a). One monocyte subpopulation (cluster 9) consisted of 98.3% of all monocytes at the severe stage, while the proportion was only 12.1% at the remission stage and 0% in healthy controls (Fig. 2b); therefore, we initially assessed these severe stage-specific monocytes.

Transcriptional differences among monocyte subtypes were detected based on a pairwise comparison of the gene expression in the severe and remission stages and respective comparisons with healthy control individuals. A large number of differentially expressed genes (DEGs) with reported inflammation-related functions were observed in the severe stage-specific monocytes, including previously reported cytokine storm-related genes such as *TNF*[8], *IL10*[8], *CCL3*[8], and *IL6*[4]; inflammation-related chemokine genes *CCL4, CCL20, CXCL2, CXCL3, CCL3L1, CCL4L2, CXCL8,* and *CXCL9*; and inflammasome activation-associated genes *NLRP3* and *IL1B* (Fig. 2c, fold change >2, $P < 10^{-3}$ according to the Wilcoxon rank-sum test; Fig. 2d; and Supplementary Data 3). Th identification of a large number of DEGs with reported inflammation-related functions supports the idea that the severe stage-specific monocyte subpopulation we detected in our single-cell COVID-19 patient data may support the development of inflammatory responses in severe COVID-19 patients.

GO analysis indicated the enrichment of genes with annotations related to regulation of the acute inflammatory response, regulation of leucocyte activation, cell chemotaxis and the cellular response to chemokines in severe-stage COVID-19 patients compared to remission-stage patients and healthy controls (Fig. 2e, f, $P < 10^{-117}$ according to the Wilcoxon rank-sum test; Supplementary Fig. 5; and Supplementary Data 4, 5), suggesting that the inflammatory storm caused by this monocyte subpopulation is suppressed by tocilizumab treatment.

Next, we explored transcription factors (TFs) in monocytes that may be involved in promoting the inflammatory storm. We used SCENIC[19] and predicted 9 TFs that may regulate genes that were upregulated in severe stage-specific monocytes (Fig. 2g). We then constructed a gene regulatory network among the TFs predicted by SCENIC and a set of inflammation-relevant genes that were collected from the literature[20,21]. We found that 3 of the SCENIC-predicted TFs, namely, *ATF3, NFIL3,* and *HIVEP2*, may have the capacity to regulate the detected inflammation-relevant genes (Supplementary Fig. 6). Additionally, we found that the expression of *ATF3, NFIL3,* and *HIVEP2* and their motif enrichment, which was predicted according to the expression of their potential target genes, were enhanced in the severe stage-specific monocyte subpopulation (Fig. 2h), further indicating that these three TFs may regulate the observed inflammatory storm in monocytes.

Recent studies have shown that over 20% of severe COVID-19 patients have symptoms of severe septic shock, which affects several organ systems and contributes to liver injury[22], acute kidney failure[23], and abnormal heart damage[24]. We therefore checked whether this severe stage-specific monocyte subpopulation is unique to patients with COVID-19. We downloaded scRNA-seq datasets from patients with sepsis at a mild stage (Int-URO) and patients with sepsis at a severe stage (ICU-SEP), as well as critically ill patients without sepsis (ICU-NoSEP) and healthy controls (Control)[25]. We then integrated these data sets with our COVID-19 patients single-cell data using Seurat[15] (version 3.1.4), which revealed a total of 10 monocyte cell clusters (Supplementary Fig. 7a, b). Interestingly, the cells from the severe stage COVID-19 patients clearly overlapped with only one of the integrated monocyte clusters (cluster VI) (Supplementary Fig. 7c), suggesting that the severe stage-specific monocyte population might be unique to COVID-19.

**A monocyte-centric cytokine/receptor interaction network.** Given that monocytes in the severe stage may be involved in the regulation of a variety of immune cell types, we used the accumulated ligand/receptor interaction database[26] CellPhoneDB (www.cellphonedb.org) to identify alterations of the molecular interactions between monocytes and all of the immune cell subsets we identified in our single-cell analysis (Supplementary Data 6). We found 15 cytokine/receptor pairs whose interactions were significantly increased in severe-stage COVID-19 patients compared to those in remission stage patients and healthy controls (Fig. 3a). It is notable that the expression of multiple inflammatory storm-related cytokines/receptors were significantly increased in severe-stage COVID-19 patients (Fig. 3b), which indicates that it is plausible that monocytes may have a substantially increased propensity for interaction with other immune cells in blood vessels. Our comparison between severe stage and remission stage patients also suggested the obvious attenuation of increased cytokine/receptor interaction activity among the immune cells of remission-stage COVID-19 patients (Fig. 3b). While they are clearly preliminary, our data support a role for tocilizumab in reducing the occurrence of monocyte receptor compositions that have been previously implicated in the induction of inflammatory storms.

Consistent with a previous report that inflammatory monocyte-released IL-6 plays a vital role in inducing an inflammatory storm in severe COVID-19 patients[4], we found that monocytes were predicted to communicate with CD4[+] T cells and plasma B cells in severe-stage COVID-19 patients through the cytokine/receptor pair IL-6/IL-6R. We also detected that the severe stage-specific monocytes showed elevated expression of other cytokine/receptor pairs that may contribute to a broad spectrum of immune cell interactions, such as TNF-α and its receptors, through which monocytes may interact with CD4[+] T, CD8[+] T, and B cells. Similarly, the severe-stage monocytes had elevated levels of IL-1β and its receptor, suggesting the potential functional interaction of these monocytes with CD8[+] T cells. Chemokines such as CCL4L2, CCL3, and CCL4 and their respective receptors were also found to be enriched in severe-stage monocytes, indicating the potential of targeting these cytokines and/or their receptors with drugs for treating severe-stage COVID-19 patients. Indeed, it is notable that inhibitors targeting some of these cytokine/receptor pairs are currently undergoing anti-COVID-19 clinical trials in multiple places around the world (Supplementary Data 7). Collectively, these findings help illustrate the possible molecular basis of cell–cell interactions in the peripheral blood of COVID-19 patients,

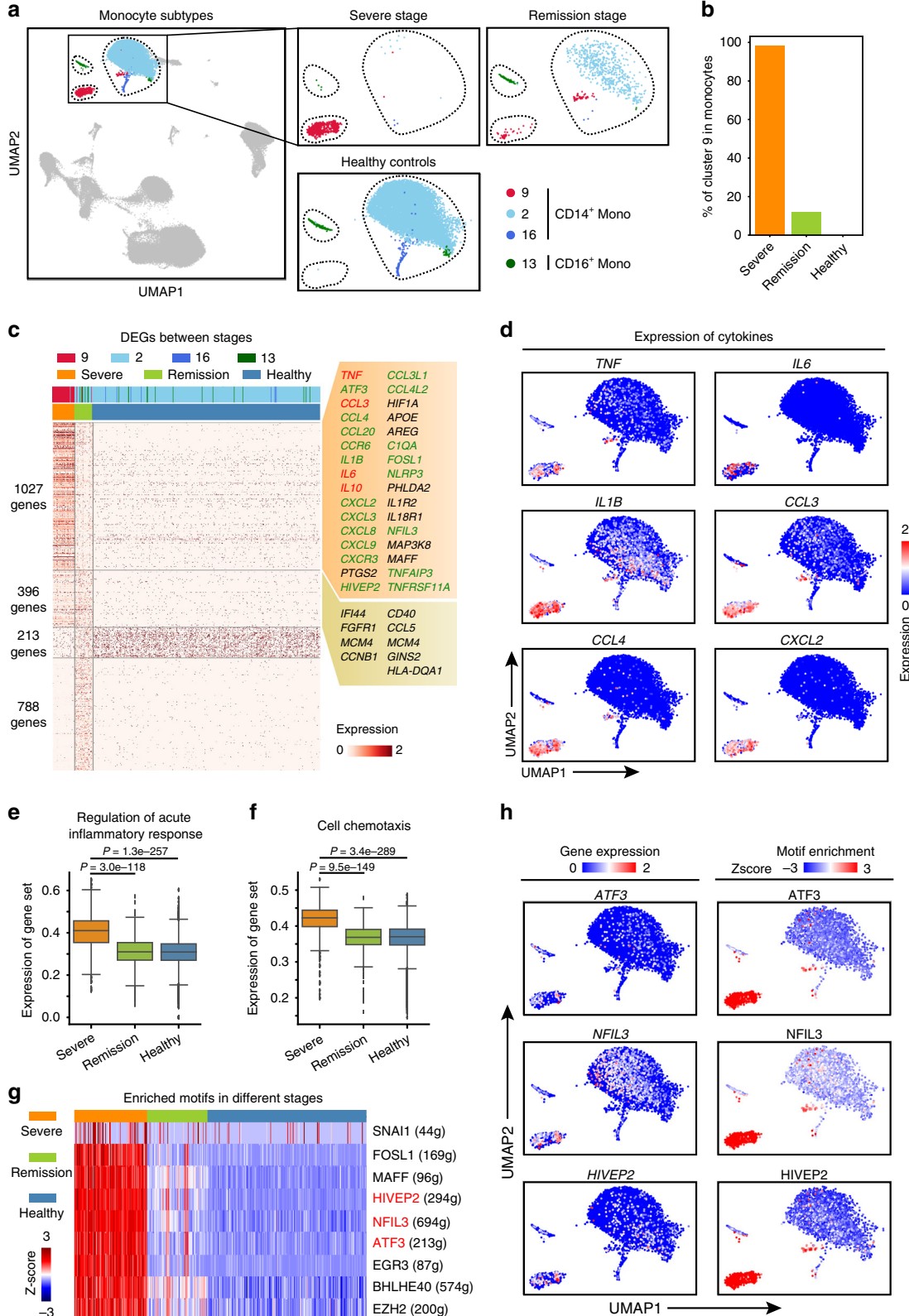

leading to a better understanding of the mechanisms of the inflammatory storm associated with the disease.

**Enhancement of humoral and cellular immune responses.** Studies of avian H7N9 disease have revealed that viral infection can elicit robust, multifactorial immune responses[27,28], and a very recent study reported effective immune responses in a nonsevere COVID-19 patient[29]. However, it is not clear whether the antiviral immune responses are affected by tocilizumab treatment. We assessed the antiviral immune responses—both humoral and cell-mediated—of severe-stage COVID-19 patients and compared them with those of both remission-stage patients and healthy controls. As expected for uninfected controls, there were hardly any plasma B cells in healthy individuals (Fig. 4a). In contrast,

**Fig. 2 A unique monocyte subpopulation contributes to inflammatory storms in severe-stage COVID-19 patients. a** UMAP plot showing three clusters of CD14+ monocytes and 1 cluster of CD16+ monocytes. Cells are colour-coded by clusters. **b** Bar plot of the proportion of monocytes in cluster 9 at the severe and remission stages and in healthy control individuals. Source data are provided as a Source Data file. **c** Heatmap of differentially expressed genes (DEGs) in monocytes from the pairwise comparison between the severe-stage patients, remission-stage patients, and healthy control individuals. **d** UMAP plots showing the expression of selected cytokines in all monocyte clusters. **e, f** Box plots of the average expression of genes involved in the signalling pathways regulation of acute inflammatory response (**e**) and cell chemotaxis (**f**) in monocytes from the severe stage ($n = 912$ cells) and remission stage ($n = 678$ cells) and in healthy control individuals ($n = 9719$ cells). Centre line, median; box limits, upper and lower quartiles; whiskers, 1.5x interquartile range; points, outliers; $P$ values were calculated using two-sided Wilcoxon rank-sum tests. Source data are provided as a Source Data file. **g** Heatmap of the area under the curve (AUC) scores of expression regulation by transcription factors (TFs), as estimated using SCENIC. The top-ranked TFs showing the highest difference in expression regulation estimates in monocytes from severe-stage COVID-19 patients are shown. **h** UMAP plots showing the expression of the *ATF3*, *NFIL3*, and *HIVEP2* genes in monocytes (top) and the AUC of the estimated regulon activity of the corresponding TFs, predicting the degree of expression regulation of their target genes (bottom).

there were many plasma B cells in both the severe- and remission-stage COVID-19 patients (Fig. 4a, b), suggesting that SARS-CoV-2 infection may elicit antiviral humoral immune responses, which are not affected by the tocilizumab treatment.

CD8+ T cells function in cell-mediated immunity against viral infections by killing infected cells and secreting proinflammatory cytokines[30]. Our single-cell analysis detected a total of 13,602 CD8+ T cells. Clustering of these cells revealed three subtypes: naïve CD8+ T cells (cluster 3), effector CD8+ T cells (cluster 6), and a subset of CD8+ T cells with expression of known proliferation markers (cluster 12) (Fig. 4c, d). The CD8+ T cells of the severe patients were primarily found in the effector CD8+ T cell cluster (Fig. 4c, d). We then conducted pairwise comparisons to identify the DEGs in the effector CD8+ T cells among the severe and remission stage patients and healthy controls (Fig. 4e, Supplementary Data 8). GO analysis indicated that the DEGs in the severe-stage effector CD8+ T cells exhibited enrichment for positive regulation of cell activation (Fig. 4f, $P < 10^{-10}$, hypergeometric test; Supplementary Data 9). In contrast, DEGs of CD8+ T cells from severe- and remission-stage COVID-19 patients (i.e., vs. healthy controls) were enriched for functional annotations related to cell chemotaxis and regulation of cell killing (Fig. 4g, $P < 10^{-6}$, hypergeometric test; Supplementary Data 9). We also detected significantly elevated expression of 306 and 94 genes associated with these GO terms (Fig. 4h, i, $P < 10^{-32}$, Wilcoxon rank-sum test; Supplementary Data 10). Together, these results indicate that SARS-CoV-2 infection elicits robust adaptive immune responses and suggest that tocilizumab treatment further promotes such responses.

To gather additional empirical support from COVID-19 patients, we downloaded the bulk RNA-seq data of PBMCs from three severe COVID-19 patients and three healthy controls[31] and applied AutoGeneS[32] to deconvolute the composition of the cell clusters based on the signature genes identified in our single-cell analysis. Our results indicated that there were significantly more severe stage-specific monocytes (cluster 9), plasma B cells (cluster 11), and proliferating CD8+ T cells (cluster 12) in severe COVID-19 patients than in healthy controls (Supplementary Fig. 8a-c, $P < 0.05$, Student's $t$ test), which are findings that are consistent with our main conclusions.

## Discussion

The immune system exerts essential functions in fighting off viral infections[33,34]. Recent studies have indicated that monocytes can exacerbate and even be a primary factor in mortality caused by COVID-19 by contributing to inflammatory storms[4]. In the present study, we used single-cell mRNA sequencing to discover a specific monocyte subpopulation that may contribute to inflammatory storms in severe-stage COVID-19 patients. By analysing monocyte-centric cytokine/receptor pairs and predicting interaction networks, we uncovered severe stage-specific

peripheral immune cell communication that may drive the inflammatory storms in COVID-19 patients. Our identification of this monocyte subpopulation and these cytokine storm-related cytokine/receptors provides mechanistic insights into the immunopathogenesis of COVID-19 and suggests the potential of these cytokine/receptor molecules as candidate drug targets for treating the disease.

There have long been questions about whether treatment with the immunosuppressive agent tocilizumab may affect the antiviral responses in the body[35,36]. The single-cell profiles illustrated a sustained humoral and cell-mediated antiviral immune response in severe- and remission-stage COVID-19 patients. For example, tocilizumab treatment in severe-stage COVID-19 patients resulted in a high proportion of plasma B cells with antibody-secreting functions and we found that the cytotoxicity and cytokine production of effector CD8+ T cells remained stable upon tocilizumab treatment.

Our work represents a collaborative clinical/basic research effort that provides an empirical method for studying single-cell resolution profiles of severe COVID-19 patients. Deconvolution analysis of published bulk RNA-seq data[31] from three additional severe COVID-19 patients and healthy controls helpd to support our conclusions on the enrichment of severe stage-specific monocytes and plasma B cells in severe-stage COVID-19 patients. We further integrated additional single-cell datasets from sepsis patients and found that the severe stage-specific monocytes we observed are unique to severe COVID-19. Based on the incorporation of a variety of additional data, our study and empirical data provide actionable insights that will help the multiple research communities who are still fighting against the virus, including clinical physicians, drug developers, and basic scientists.

## Methods

**Human samples.** Peripheral blood samples were obtained from two severe COVID-19 patients. Patient severity was defined according to the Diagnosis and Treatment of COVID-19 (Trial Version 6) which was released by The General Office of the National Health Commission and the Office of the National Administration of Traditional Chinese Medicine. Patient P1 was defined as a severe patient because his peripheral capillary oxygen saturation (SPO2) was <93% without a nasal catheter supplying oxygen. Patient P2 was defined as having critical illness due to respiratory failure, multiple organ dysfunction (MOD), and a SPO2 < 93 without a nasal catheter supplying oxygen. Two peripheral blood samples were obtained from patient P1 on day 1 and day 5, and three peripheral blood samples were obtained from patient P2 on day 1, day 5, and day 7. For both patients, the peripheral blood samples for day 1 were collected within 12 h of tocilizumab administration, when the patients were still in the severe stage. Our decision to obtain blood from the two patients on day 5 was guided by information from the authors of the recent study published in *PNAS*[13], which prompted our decision to consider day 5 of tocilizumab treatment as the beginning of the remission stage. For patient P2, we observed that a SARS-CoV-2 nucleic acid test of a throat swab specimen was still positive on day 5, so we performed another blood draw on day 7 for P2, at which point the throat swab specimen nucleic acid test was negative. All samples were collected from the First Affiliated Hospital of the University of Science and Technology of China. Before the blood draws, informed consent was

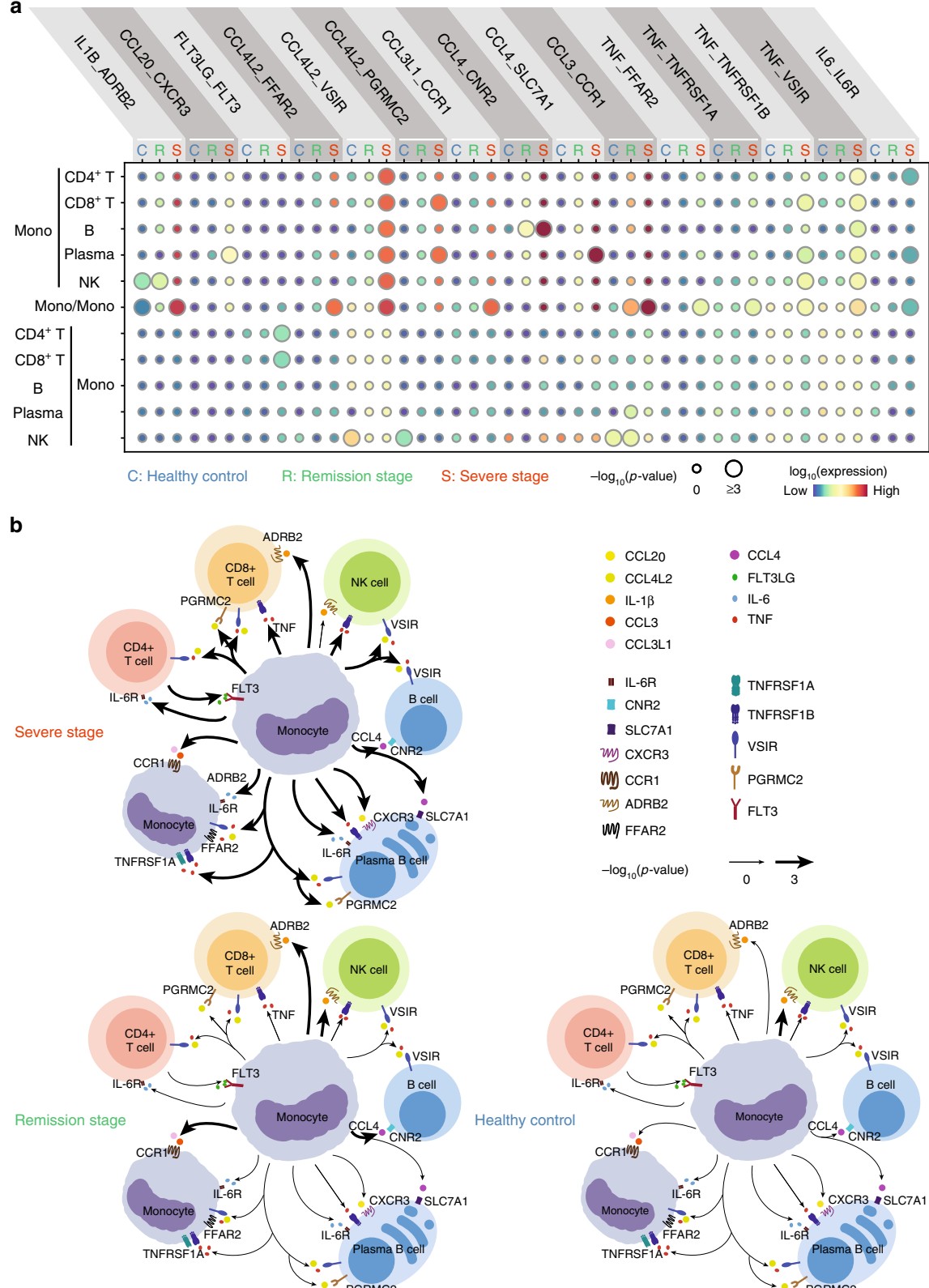

**Fig. 3 The monocyte-centric molecular interactions of peripheral immune cells in severe-stage COVID-19 patients. a** Dot plot of the predicted interactions between monocytes and the indicated immune cell types in the severe and remission stages and in healthy control individuals. *P* values are indicated by the circle sizes, as shown in the scale on the right (permutation test). The means of the average expression level of interacting molecule 1 in cluster 1 and interacting molecule 2 in cluster 2 are indicated by the colour. Assays were carried out at the mRNA level but were extrapolated to protein interactions. **b** Summary illustration depicting the potential cytokine/receptor interactions between monocytes and other types of peripheral immune cells in the severe and remission stages and in healthy control individuals. Bolded lines indicate predicted enriched cytokine/receptor interactions between monocytes and other immune cell types.

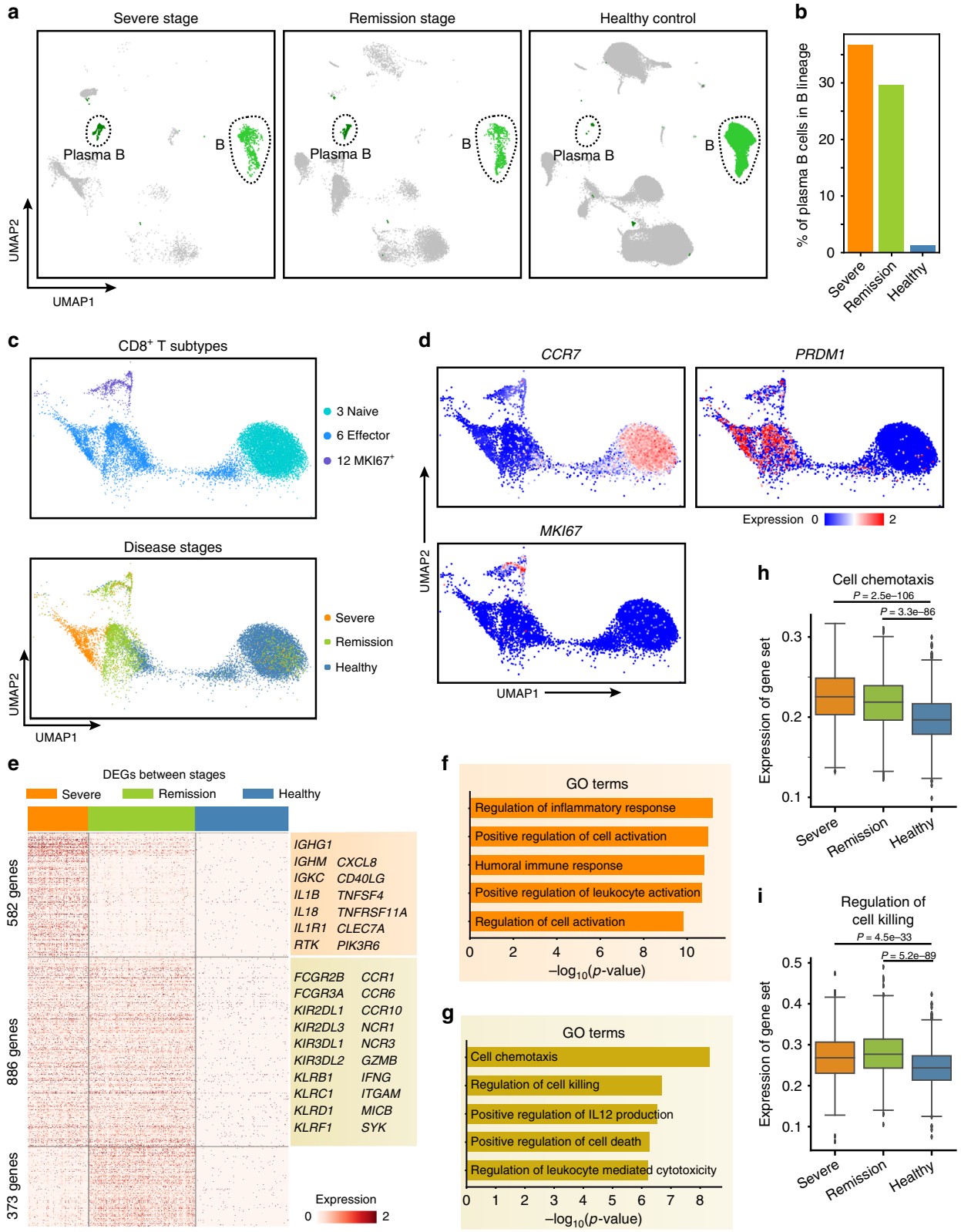

obtained from each patient. Ethical approval was obtained from the ethics committee of the First Affiliated Hospital of the University of Science and Technology of China (No. 2020-XG(H)-020).

**Cell isolation.** We collected 2 ml peripheral blood each time from the COVID-19 patients. PBMCs were freshly isolated from whole blood by using a density gradient centrifugation with Ficoll-Paque and cryopreserved for the subsequent generation of a single-cell RNA library.

**Single-cell RNA-seq.** We generated single-cell transcriptome libraries by following the instructions of the single-cell 3′ solution v2 reagent kit (10X Genomics). Briefly, after thawing, washing, and counting cells, we loaded the cell suspensions onto a chromium single-cell chip along with partitioning oil, reverse transcription (RT) reagents, and a collection of gel beads that contained 3,500,000 unique 10X barcodes. After the generation of single-cell gel bead-in-emulsions (GEMs), RT was performed using a C1000 Touch™ Thermal Cycler (Bio-Rad). The amplified cDNA was purified with SPRIselect beads (Beckman Coulter). Single-cell libraries

**Fig. 4 Enhanced humoral and cell-mediated immunity in severe COVID-19 patients. a** UMAP representations of B cell and plasma B cell clusters from the severe and remission stages and in healthy control individuals. **b** Bar plot of the proportions of plasma B cells in the B cell lineage from the severe and remission stages and in healthy control individuals. Source data are provided as a Source Data file. **c** UMAP representations of CD8+ T cell subtypes (left) and the distribution of cells from the severe and remission stages and in healthy control individuals in each subtype (right). **d** Dot plot of the expression of the *CCR7*, *PRDM1*, and *MKI67* genes in all CD8+ T cell subtypes. **e** Heatmap of differentially expressed genes in effector CD8+ T cells based on pairwise comparisons between severe-stage patients, remission-stage patients, and healthy control individuals. **f, g,** Bar plots of GO terms enriched in effector CD8+ T cells from the severe stage (**f**) or the severe and remission stages (**g**). *P* values were calculated using a hypergeometric test and the Benjamini–Hochberg correction algorithm (i.e. multi-test adjustments) in Metascape. **h, i,** Box plots of the average expression of genes involved in the signalling pathways associated with cell chemotaxis (**h**) and regulation of cell killing (**i**) in effector CD8+ T cells from the severe stage ($n = 1128$ cells) and remission stage ($n = 1964$ cells) and in healthy control individuals ($n = 1715$ cells). Centre line, median; box limits, upper and lower quartiles; whiskers, 1.5x interquartile range; points, outliers; *P* values were calculated using two-sided Wilcoxon rank-sum tests. Source data are provided as a Source Data file.

were then constructed following fragmentation, end repair, polyA tailing, adaptor ligation, and size selection based on the manufacturer's standard parameters. Each sequencing library was generated with a unique sample index. Libraries were sequenced on the Illumina NovaSeq 6000 system.

**Single-cell RNA-seq data processing.** The raw sequencing data of patients and healthy donors were processed using Cell Ranger (version 3.1.0) against the GRCh38 human reference genome with default parameters, and data from different patients and disease stages were combined with the Cell Ranger aggr function. We have uploaded the scRNA-seq data of the PBMCs from the two severe COVID-19 patients to the Genome Sequence Archive for Human (GSA-Human) at the BIG Data Centre. We also used the scRNA-seq data of PBMCs from two healthy donors, which can be downloaded from the 10X genomics official website. First, we filtered low-quality cells using Seurat[15] (version 3.1.4). For cells from COVID-19 patients (P1 and P2), we retained cells with detected gene numbers between 500 and 6000 and less than 10% mitochondrial unique molecular identifiers (UMIs). For cells from healthy donors, we retained cells with detected gene numbers between 300 and 5000 and less than 10% mitochondrial UMIs. Subsequently, we used Scrublet[14] (version 0.2.1) to eliminate doublets among the PBMCs from the COVID-19 patients and healthy donors. We used the default parameters for Scrublet (i.e., Eq. min_gene_variability_pctl = 85, n_prin_comps = 30, threshold = 0.25) and detected 50 doublets in the patients and 997 doublets in the healthy donors. After removing the doublets, we normalized the gene counts for each cell using the NormalizeData function of Seurat with the default parameters.

For the downstream data processing, we used canonical correlation analysis and the top 40 canonical components to identify the anchor cells in patients and healthy controls. We then used the IntegrateData function in Seurat to integrate the cells from COVID-19 patients and healthy controls. We clustered all the cells based on the integrated gene expression matrix using Seurat with a parameter Resolution=0.3 and generated 20 clusters. To display the cells in a two-dimensional space, we performed principal component analysis on the integrated dataset and used the first 50 principal components (PCs) for uniform manifold approximation and projection (UMAP) analysis.

In the integration of cells from COVID-19 and sepsis patients using Seurat, we applied the same functions and parameters as those described above. We used Seurat to cluster the integrated gene expression matrix (with a resolution = 0.3) and identified monocyte clusters based on the expression of the known marker genes *CD14* and *CD68*. We then extracted all monocytes from the integrated dataset and reclustered them. Finally, we generated 10 cell clusters.

**Integration analysis with harmony.** To verify the reliability of the integration results obtained using Seurat (version 3.1.4), we also applied Harmony[17] to integrate the PBMCs from COVID-19 patients and healthy controls. We used the same gene expression matrix and applied the same parameters as those used in Seurat and used the first 50 PCs to perform the data integration by calling the RunHarmony function in Harmony. We then used the same clustering algorithm as that used in Seurat to cluster the cells and generated 23 clusters (resolution = 0.5) based on the integration results obtained from Harmony. The Jaccard index was applied to gauge the similarity between the cell clusters, and the cell integration was processed by Seurat or by Harmony. The Jaccard similarity $J(C_i, C_j)$ between each pair of Seurat clusters (cluster $i$) and Harmony clusters (cluster $j$) was defined as follows:

$$J\left(C_i, C_j\right) = \frac{|C_i \cap C_j|}{|C_i \cup C_j|},\quad(1)$$

where $C_i$ and $C_j$ are the cells belonging to clusters $i$ and $j$, respectively.

**Differential expression analysis.** To search for the DEGs, we first set the negative elements in the integrated expression matrix to zero. We used the Wilcoxon rank-sum test to search for the DEGs between each pair of cells obtained from the three groups (i.e., the severe stage, remission stage, and healthy control groups). We applied multiple thresholds to screen for DEGs, including a

mean fold change >2, a *P* value < 0.001 and detection in >10% of cells in at least one stage.

We defined stage A specific-DEGs according to the intersections between the DEGs in stage A versus those in stage B and the DEGs in stage A versus those in stage C. We defined the stage A and B shared DEGs according to the intersections of the DEGs in stage A versus those in stage C and the DEGs in stage B versus those in stage C minus the DEGs between stage A and B. In this way, we obtained the specific DEGs for each stage and the shared DEGs for each pair of the 3 stages. We then uploaded these DEG groups to the Metascape[37] website (https://metascape.org/gp/index.html#/main/step1) and used the default parameters to perform Gene Ontology (GO) analysis for each stage.

**Motif enrichment and regulatory network.** We used SCENIC[19] (version 1.1.2) and the RcisTarget database to build the gene regulatory network of CD14+ monocytes. Since the number of CD14+ monocytes from healthy controls (N = 9,618) was greater than that from the severe- and remission-stage patients (N = 1,607), to balance their contributions to the motif analysis, we randomly sampled 2000 CD14+ monocytes from the healthy controls for calculation. We selected 13,344 genes that were detected in at least 100 monocytes or included among the DEGs of the three stages as the input features for SCENIC. With the default parameters, SCENIC generated the enrichment scores of 427 motifs. We used Student's *t* test to calculate the *P* values of these motifs when compared between severe-stage patients and healthy controls and selected the severe-specific enriched motifs with a fold change >1.5 and a *P* value < $10^{-100}$.

We then inputted the enrichment scores of the severe-specific enriched motifs and the expression of their targeted genes into Cytoscape[38] to construct a connection map for the gene regulatory network, as shown in Supplementary Fig. 6. The thickness of the line connecting TFs and target genes represented the weight of the regulatory link predicted by SCENIC.

**Cytokine/receptor interaction analysis.** To identify potential cellular communications between monocytes and other cell types (CD4+ T, CD8+ T, B, plasma B, and NK cells), we applied the CellphoneDB[26] algorithm to the scRNA-seq profiles from the severe and remission stages and healthy control individuals. CellphoneDB evaluated the impact of the ligand/receptor interactions based on ligand expression in one cell type and the corresponding receptor expression in another cell type. We focused on the enriched cytokine/receptor interactions in severe-stage COVID-19 patients and selected the cytokine/receptor interactions associated with highly significant (*P* value < 0.05) cell–cell interaction pairs in the severe stage compared to the remission stage and healthy individuals. We also included cytokine/receptor pairs that were highly expressed in the severe stage.

**Deconvolution of cell clusters from bulk RNA-seq data.** We applied Auto-GeneS[32] to deconvolute the composition of the cell clusters based on the signature genes identified in our single-cell analysis. Specifically, we first obtained a gene-by-cluster expression matrix from our normalized single-cell profile, in which the matrix elements represented the average expression of each gene in each cell cluster. We then defined the top 5000 most variable genes between the cell clusters and the 2000 DEGs used for cell clustering as VarGenes and extracted the VarGenes-by-cluster expression matrix as the feature gene expression profile for AutoGeneS. We set the input parameters as model = nusvr, ngen = 1000, seed = 0, and nfeatures = 1500 to deconvolute the cell composition in AutoGeneS.

**Statistical analysis.** The two-tailed Wilcoxon rank-sum test (also called the Mann-Whitney U test) was used to identify the DEGs and to compare the differences in the expression of genes of interest between two conditions. In CellphoneDB, a permutation test was used to evaluate the significance of a cytokine/receptor pair. Metascape utilizes the hypergeometric test and the Benjamini-Hochberg *P* value correction algorithm to identify the ontology terms that are associated with a significantly greater number of shared genes than expected. We used Student's *t* test to evaluate the significance of the expression differences of

the TFs (and their target genes) between samples from severe-stage patients and healthy controls.

**Reporting summary**. Further information on research design is available in the Nature Research Reporting Summary linked to this article.

## Data availability

The scRNA-seq data of PBMCs from the 2 severe COVID-19 patients can be obtained from the Gene Expression Omnibus (GEO) database with the accession number GSE150861. We also used published datasets as controls or comparable data, including (1) the scRNA-seq data of PBMCs from 2 healthy donors downloaded from the 10X Genomics official website [https://support.10xgenomics.com/single-cell-gene-expression/datasets/3.1.0/5k_pbmc_NGSC3_aggr]; (2) the scRNA-seq data of PBMCs from 22 sepsis patients and 19 related controls[25], which is available on the Institute Single Cell Portal [https://singlecell.broadinstitute.org/single_cell] under accession number SCP548; (3) the bulk RNA-seq data of PBMCs from 3 COVID-19 patients and 3 related controls[31], which were downloaded from the GSA at the BIG Data Centre under accession number CRA002390; and (4) the GRCh38 human reference genome used for the sequencing data alignment, which is available on the 10X Genomics official website [https://support.10xgenomics.com/single-cell-gene-expression/software/downloads/latest]. Source data are provided with this paper. Source data are provided with this paper.

## Code availability

The analysis scripts are accessible at Github: https://github.com/QuKunLab/COVID-19. Source data are provided with this paper.

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

## Acknowledgements

This work was supported by the National Key R&D Program of China (2017YFA0102900 to K.Q.), the National Natural Science Foundation of China grants (91940306, 81788101, 31970858, 31771428 and 91640113 to K.Q., 31700796 to C.G. and 81871479 to J.L.), the Fundamental Research Funds for the Central Universities (YD2070002019 and WK2070000158 to K.Q.). We thank the USTC supercomputing center and the School of Life Science Bioinformatics Center for providing super-computing resources for this project. We thank the CAS interdisciplinary innovation team for helpful discussion.

## Author contributions

K.Q. conceived and supervised the project; K.Q., C.G., and J.L. designed the experiments; C.G. and J.L. performed the experiments and conducted all the sample preparation for next-generation sequencing with the help from H.M. and T.J.; B.L. performed the data analysis with the help from P.C., Q.Y., L.Z., L.J., C.J., Q.L., D.Z., W.Z., Y.L., K.L., X.G., and J.F.; T.J., X.W., L.L., J.W., and X.M. provided COVID-19 blood samples and clinical information; J.W. contributed to the revision of the manuscript; K.Q., C.G., J.L., and B.L. wrote the manuscript with the help of B.F., H.W. and all the other authors.

## Competing interests

Jingwen Fang is the chief executive officer of HanGen Biotech. The other authors declare no competing interests.
