## [Peer Review File · Nature Communications]

REVIEWERS' COMMENTS:

Reviewer #1 (Remarks to the Author):

The manuscript has improved significantly following revision. My remaining concern is reflected in the title.

This is a descriptive study and in no way dissects the mechanism of inflammation seen in COVID-19 peripheral blood compartment. The authors describe an inflammatory monocyte population that are observed in these patients but there is no evidence that these inflammatory monocytes drive the inflammatory storm. There is also no evidence that Tocilizumab caused the attenuation at later stage of the illness. The title, summary and manuscript content need to reflect this clearly e.g. Single-cell analysis of severe COVID-19 patients reveals inflammatory monocytes. It is difficult to say from n=2 that Tocilizumab was responsible for any of the changes observed over time.

I would also suggest the authors avoid the phrase 'remission stage' but simply use day 5-7 post treatment as remission implies some benefit for treatment which is hard to justify given the two case studies presented.

Response to reviewers

We thank the reviewer for his/her positive assessment of our revision and further suggestions for improving our study. We have now provided our point-by-point responses as below:

Reviewer #1 (Remarks to the Author):

(1) The manuscript has improved significantly following revision. My remaining concern is reflected in the title.

Response: We thank the reviewer for the positive assessment of our revision. Guided by the editor's recommendation, we have now changed the title of our manuscript to "Single-cell analysis of two severe COVID-19 patients reveals a monocyte-associated and tocilizumab-responding cytokine storm".

(2) This is a descriptive study and in no way dissects the mechanism of inflammation seen in COVID-19 peripheral blood compartment. The authors describe an inflammatory monocyte population that are observed in these patients but there is no evidence that these inflammatory monocytes drive the inflammatory storm. There is also no evidence that Tocilizumab caused the attenuation at later stage of the illness. The title, summary and manuscript content need to reflect this clearly e.g. Single-cell analysis of severe COVID-19 patients reveals inflammatory monocytes. It is difficult to say from n=2 that Tocilizumab was responsible for any of the changes observed over time.

Response: We thank the reviewer for helping us to re-consider our claims in light of this important concern. We appreciate how the two case studies could influence any conclusions we draw and we do not have strong evidence that these inflammatory monocytes drive the inflammatory storm, therefore we have taken special care in the revised manuscript to avoid using imprecise phrases.

We have provided all available clinical information which support that both patients were at a severe disease stage on day 1, and their symptoms were tending to normal on day 5 and day 7 (Supplementary Table 1). Besides, a recent study published in *National Science Review*¹ showed that the inflammatory monocytes incite inflammatory storms in severe COVID-19 patients, another study published in *PNAS*² revealed that the clinical symptoms of severe COVID-19 patients were remarkably improved on day 5 after tocilizumab treatment. In combination with our discoveries, these data demonstrate that tocilizumab is responsible for the attenuation of cytokine storm in severe COVID-19 patients.

In response, we have changed the title of our manuscript to "Single-cell analysis of two severe COVID-19 patients reveals a monocyte-associated and tocilizumab-responding cytokine storm", and revised the **Abstract** of our manuscript guided by the editor's recommendation.

(3) I would also suggest the authors avoid the phrase 'remission stage' but simply use day 5-7 post treatment as remission implies some benefit for treatment which is hard to justify given the two case studies presented.

Response: We thank the reviewer for focusing our attention on this important concern. A very recent study published in *PNAS*² showed that the clinical symptoms of severe COVID-19 patients were remarkably improved on day 5 after tocilizumab treatment. Specifically, on the 5th day after treatment, the peripheral oxygen saturation improved in 75.0% patients (15/20); CRP decreased significantly and returned to normal in 84.2% patients (16/19); CT scans showed that the lesions were absorbed in 90.5% patients (19/21); white blood cell counts returned to normal in 89.5% patients (17/19), suggesting an efficient therapeutic with tocilizumab, with patients given fewer than 5 days of this treatment likely to be in an early remission stage. We do understand that our patient number is low, however our decisions to draw blood on day 5 to represent a remission stage was based on the clinical status of the two patients, as well as our interactions with the authors of a recent publication about tocilizumab therapy for COVID-19², which was reported in a cohort of 20 severe COVID-19 patients. Therefore, we suggest to retain the phrase of 'remission stage' to describe the clinical status of the patients when samples were collected on day 5 or day 7.

References:

- 1 Zhou, Y. G. *et al.* Pathogenic T-cells and inflammatory monocytes incite inflammatory storms in severe COVID-19 patients. *Natl Sci Rev* **7**, 998-1002, doi:10.1093/nsr/nwaa041 (2020).
- 2 Xu, X. *et al.* Effective treatment of severe COVID-19 patients with tocilizumab. *Proc Natl Acad Sci U S A* **117**, 10970-10975, doi:10.1073/pnas.2005615117 (2020).